# SARS-CoV-2 Pre-Exposure Prophylaxis with Sotrovimab and Tixagevimab/Cilgavimab in Immunocompromised Patients—A Single-Center Experience

**DOI:** 10.3390/v14102278

**Published:** 2022-10-17

**Authors:** David Totschnig, Max Augustin, Iulia Niculescu, Hermann Laferl, Sonja Jansen-Skoupy, Clara Lehmann, Christoph Wenisch, Alexander Zoufaly

**Affiliations:** 1Department of Medicine IV, Klinik Favoriten, Vienna Healthcare Group, Kundratstraße 3, 1100 Vienna, Austria; 2Department I of Internal Medicine, Medical Faculty and University Hospital Cologne, University of Cologne, 50937 Cologne, Germany; 3Department of Laboratory Medicine, Klinik Favoriten, Vienna Healthcare Group, Kundratstraße 3, 1100 Vienna, Austria

**Keywords:** SARS-CoV-2 PrEP, pre-exposure prophylaxis, immunocompromised patients, sotrovimab, tixagevimab/cilgavimab

## Abstract

Immunocompromised patients experience reduced vaccine effectiveness and are at higher risk for coronavirus disease 19 (COVID-19) death. Pre-exposure prophylaxis (PrEP) aims to protect these patients. So far, only tixagevimab/cilgavimab is authorized for use as PrEP. This paper aims to provide real-world data on the use of tixagevimab/cilgavimab and sotrovimab as severe acute respiratory syndrome coronavirus 2 (SARS-CoV-2) PrEP in immunocompromised patients, comparing the evolution of antibody levels and reporting the incidence of breakthrough infections. A retrospective, single-center analysis was conducted including 132 immunocompromised patients with inadequate vaccine response, who received COVID-PrEP at our clinic between January and June 2022. Initially, 95 patients received sotrovimab while 37 patients received tixagevimab/cilgavimab. Antibody levels after first PrEP with sotrovimab remain high for several months after infusion (median 10,058 and 7235 BAU/mL after 1 and 3 months, respectively), with higher titers than after tixagevimab/cilgavimab injection even 3 months later (7235 vs. 1647 BAU/mL, *p* = 0.0007). Overall, breakthrough infections were rare (13/132, 10%) when compared to overall infection rates during this period (over 30% of the Austrian population), with mild disease course and rapid viral clearance (median 10 days). Sotrovimab may be an additional option for SARS-CoV-2 PrEP.

## 1. Introduction

In December 2019, a novel coronavirus, termed severe acute respiratory syndrome coronavirus 2 (SARS-CoV-2), was first described in Wuhan, China [1], causing coronavirus disease 19 (COVID-19). It has since spread across the world leading to a global health crisis with over 500 million cases and 6.2 million deaths reported thus far [2].

Since their introduction in early 2021, SARS-CoV-2-vaccines have been a crucial tool in preventing infection and severe disease [3,4]. Immunocompromised patients, however, experience reduced vaccine effectiveness [5,6,7] and are at higher risk for COVID-19 death [8]. They show impaired seroconversion after vaccination, which is especially poor in organ transplant recipients and patients with hematological malignancies [9]. For instance, Herishanu et al. found 1315.5 times lower antibody titers among 52 patients with chronic lymphocytic leukemia (median: 0.824 BAU/mL; IQR: 0.4 to 167.3 BAU/mL) as compared to 52 healthy control subjects (median 1084 BAU/mL; IQR: 128.9 to 1879 BAU/mL) after the second dose of BNT162b2 mRNA COVID-19 vaccine [10]. Other causes of immunosuppression include treatment for autoimmune and rheumatologic diseases, chemotherapy, and genetic immunodeficiencies.

Recently, a new therapeutic option emerged for these patients: based on the results of the PROVENT trial, the FDA issued an Emergency Use Authorization for the long-acting monoclonal antibodies (mAbs) tixagevimab and cilgavimab (TIX/CIL) for pre-exposure prophylaxis (PrEP) of COVID-19 [11,12]. These antibodies have a modified Fc receptor to increase their half-life to about 90 days [13]. They can thus be administered every 6 months to patients at risk for inadequate response to active immunization, and were shown to reduce the risk of developing symptomatic COVID-19 by 77% compared to placebo [12]. Sotrovimab (SOT) is a mAb derived from S309, an antibody isolated from a SARS-CoV-1 survivor, which recognizes an epitope containing a glycan that is conserved within the Sarbecovirus subgenus [14,15]. It also contains a modified Fc receptor, extending its half-life to 50 days. SOT is used for the treatment of early COVID-19 and is currently being investigated for PrEP [16,17,18]. An important aspect is the potential role of immunosuppressed patients in the emergence of new variants, which PrEP might help to prevent [19].

A significant challenge to the use of mAbs is the emergence of new SARS-CoV-2 variants of concern (VOCs). Since being first reported in November 2021, the SARS-CoV-2 Omicron VOC has rapidly become predominant in most countries [20], with several sublineages emerging since then. Notably, in early 2022, the BA.2 sublineage largely replaced the original Omicron strain (BA.1), and was then itself replaced by the BA.4/5 sublineages [21]. The most recent sublineage, BA.2.75, first identified in May 2022, has shown a growth advantage over BA.5 in India and may eventually become the dominant strain [22]. In vitro neutralization testing on these variants shows significantly impaired affinity for most clinically available mAbs. For instance, casirivimab/imdevimab and adintrevimab are unable to neutralize any current Omicron variants. SOT and TIX/CIL have impaired affinity to all Omicron variants, with a 13-fold and 25-fold reduction in the neutralization of the BA.2.75 variant, respectively [23,24]. Only bebtelovimab currently retains in vitro potency against all Omicron sublineages, but its half-life is only 10 days [24,25,26]. It should be noted, however, that in vivo data on mAb effectiveness for the different variants are scarce, and it is unclear to what degree the reduced in vitro neutralization translates into reduced in vivo effectiveness [26,27]. Moreover, these susceptibilities are prone to further changes with the emergence of new variants, and previously discarded mAbs may become effective again in the future.

Here we present data on immunocompromised outpatients who were followed up for incident SARS-CoV2 infections after receiving SOT or TIX/CIL as PrEP.

## 2. Methods

From January 2022, immunocompromised patients without signs of SARS-CoV2 infection and without adequate response to previous vaccinations were evaluated at our outpatient department for potential prophylactic strategies, including PrEP with monoclonal antibodies, and followed up for incident SARS-CoV2 infections. Inadequate vaccination response was initially defined as antibody titers below 264 BAU/mL [28], although exceptions were made if deemed appropriate by the treating physician due to the severity of the immunosuppression or underlying disease. The patients included in this study had previously received a mean of 3.7 vaccinations, resulting in a median antibody value of 9 BAU/mL. Patients were referred from various specialized clinics in Vienna. We initially administered SOT 500 mg off-label with infusions every 2 months and performed monthly antibody testing. At each visit, patients were asked about suspected or confirmed SARS-CoV-2 infection in the past months. We also inquired about infection status during routine scheduling calls. In mid-March, we switched to TIX/CIL 150 mg/150 mg due to improved neutralization of the Omicron BA.2 variant. From this point onward, all patients received TIX/CIL regardless of previous administration of sotrovimab (Figure 1), with antibody testing every 2 months.

Antibody measurements were performed with the “Elecsys Anti-SARS-CoV-2 S (Roche, Switzerland) [29]” assay, which measures antibodies against the receptor-binding domain of the SARS-CoV-2 spike protein. Results are equivalent to the World Health Organization (WHO) standard (Roche units (U)/mL = binding antibody units (BAU)/mL), with a correlation factor of 0.9996. The positive predictive value for the presence of neutralizing antibodies is 96.27% for a cut-off value > 0.8 U/mL and 99.1% for a cut-off > 15 U/mL. According to the manufacturer’s instructions, samples with anti-SARS-CoV-2 S concentrations above the measuring range (0.4–250 U/mL) have been diluted up to 1:100. Higher results were reported as >25,000 U/mL. This capping affected some values at the T0 +1 h time point. For graphical representation, these values >25,000 were approximated as 25,000 U/mL. We performed a retrospective chart review of all patients having received mAbs for COVID-19 PrEP from 1 January to 1 June 2022. We recorded the date of all infusions and antibody-measurements, and the incidence of breakthrough SARS-CoV-2 infections. Breakthrough infections were confirmed by PCR testing. Follow-up PCR testing was done according to the recommendations of the local authorities, usually on days 5 and 10. Data were analyzed using GraphPad Prism 9 (Dotmatics, Boston, MA, USA). Mann–Whitney U test was used to compare median antibody levels. Ethics approval was obtained before initiation of the study.

## 3. Results

### 3.1. Patient Characteristics

Patient characteristics are shown in Table 1. The most common cause for immunosuppression was hematologic malignancy, followed by various autoimmune diseases, multiple sclerosis, genetic immunodeficiency, and organ transplantation. The most common immunosuppressive medications by far were anti-CD20 mAbs such as rituximab. Patients had been vaccinated against SARS-CoV-2 a mean of 3.7 times, mostly with mRNA vaccines, before the first PrEP, resulting in a median of 9 BAU/mL antibodies. First PrEP was mostly done with SOT, since TIX/CIL only became available mid-March. Follow-up PrEP was mostly done with TIX/CIL due to the improved neutralization of the BA.2 variant.

To improve clarity and readability, follow-up visits were grouped by time points, roughly equal to the number of months since the first mAb infusion. Most patients adhered to monthly visits, when we performed antibody testing and, if deemed appropriate, follow-up PrEP. Visits between follow-up days 0 and 44 were allocated to time point T1, days 45–74 to T2, and days 75–104 to T3. This resulted in T1 occurring at a mean of 30.4 (±3.1) days, T2 at 59.5 (±6.7) days, and T3 at 91.3 (±6.8) days. In the following results and discussion, these time points are used interchangeably with “after X months” to improve readability.

### 3.2. Antibody Measurements

Antibody measurements were performed 1 h after administration for SOT only due to the delayed release of TIX/CIL. Further measurements were performed monthly from January to March, and every other month starting in April. Follow-up PrEP with TIX/CIL was administered starting mid-March to all patients, due to availability and better neutralization of the BA.2 variant. This made the interpretation of subsequent antibody measurements difficult since values are heterogeneous due to varying previous infusions with SOT. Thus, we analyzed the antibody levels collected after the first PrEP only, and only before the administration of any further PrEP (Figure 2A,B). Of the 132 patients, 116 were included in the antibody analysis, while 16 were excluded due to incomplete data or lack of follow-up (Figure 1).

Among these 116 patients, median antibody values 1, 2, and 3 months after SOT were 10,058, 8160, and 7235 BAU/mL, respectively. Median antibody values 1 and 3 months after TIX/CIL were 3965 (*p* < 0.0001) and 1647 (*p* = 0.0007) BAU/mL, respectively. Antibody values after 3 months were only available for 10 SOT and 5 TIX/CIL patients at the time of writing.

### 3.3. Breakthrough Infections

Breakthrough infection (BI) occurred in 13 of 132 patients (10%) (Table 2 and Figure 3), a median of 40 days after the last PrEP. Among these, 11/13 BIs occurred after SOT PrEP and 2/13 after TIX/CIL PrEP. Notably, however, most BIs occurred in February and March, during the peak of the Omicron BA.2 wave in Austria before TIX/CIL was being administered. Patients with BI had a mean age of 60.2, and 9/13 were male. Patients with hematologic malignancy had the highest rate of BI (13.8%), but only 3 out of the 13 BIs had received anti-CD20 treatment. The most frequent symptoms were coughing, sore throat, and fatigue, with one patient being symptomatic. No patient reported dyspnea. Hospitalization occurred in one patient. The patient never required supplemental oxygen and was discharged 2 days later. The median duration until the resolution of symptoms was 4 days, median duration until the SARS-CoV-2 swab with a cycle threshold (Ct) value > 30 (without dropping below 30 again later) was 10 days. Notably, two patients received nirmatrelvir/ritonavir as antiviral outpatient therapy for 5 days.

## 4. Discussion

This retrospective analysis aimed to provide real-world data on the use of SOT and TIX/CIL for SARS-CoV-2 PrEP in immunocompromised patients.

With regard to the patient characteristics, the high prevalence of anti-CD20 therapy (Table 1) highlights the impact of these drugs on seroconversion after the vaccine, as has previously been shown [29]. While the T-cell response still occurs [30,31], overall vaccine efficacy is reduced [7] and severe disease is frequent [32]. These treatments must thus undergo a strict risk–benefit assessment for each patient for the duration of this pandemic.

Our study provides real-world pharmacological data on the off-label use of SOT as PrEP. Initial antibody levels are high and remain so for several months (Figure 2A,B), with median values after 3 months still higher than those achieved with TIX/CIL in standard dosage (150/150 mg) after 1 month. While follow-up data is only available for up to 3 months, the longer half-life of 90 days for TIX/CIL compared to 50 days for SOT suggests that antibody levels after TIX/CIL will eventually be higher if no re-dosing is performed. The difference in antibody levels can partially be explained by the difference in dosage (SOT 500 mg vs. TIX/CIL 150/150 mg) and the different mode of application (intravenous for SOT vs. intramuscular for TIX/CIL), with delayed release of TIX/CIL. Furthermore, the clinical significance of these quantitative antibody levels is uncertain.

Studies after vaccination estimate cut-off values for efficient protection ranging from 154 to 2000 BAU/mL [33,34,35,36], but they are mostly based on data derived from SARS-CoV-2 wildtype, and the role of T-cell response is unclear. 

The “Tissue-Distribution Adjusted EC90 Approach“ is an attempt to extrapolate clinical effectiveness from serum antibody concentration, by estimating the number of antibodies in the epithelial lining fluid of the lungs and comparing it to in vitro 90% effective concentration (EC90) values. Using this approach for SOT with a conservative lung-to-serum ratio of 6.5% [37], the EC90 value of 1.2 nM (=24 U/mL using the Elecsys assay) as listed by the manufacturer [38], and a reduced neutralization by a factor of 13 for BA.2.75 [24], and an estimated serum concentration of 4800 U/mL are required in the Elecsys assay to reach SOT EC90 values against BA.2.75 in the epithelial lining fluid of the lung. This value is significantly lower than the median value 3 months after SOT PrEP of 7235 U/mL described in our study (Figure 2B).

This approach, however, has several limitations. Firstly, the lung-to-serum ratio differs between mAbs, and available information is heterogeneous. For SOT, the center for Drug Evaluation and Research estimates a lung-to-serum ratio of 6.5–12% [37], while the EMA suggests 25% [14]. Furthermore, there is a number of different neutralization assays employed to estimate effectiveness against new variants. Some employ live viruses, others use pseudotyped viruses, and there are various cell lines, virus inoculum sizes, and assay durations, leading to significant disparities between results [39]. Complicating matters further, there is a multitude of potential mechanisms by which antibodies can be effective, which include direct blocking of viral entry, mAb-mediated effector functions, indirect blocking of viral entry by cross-linking, preventing the exit of virus from infected cells, and blocking cell-to-cell spread [40]. The importance of these functions may differ depending on the mAb’s target, and it is unclear to what degree each of these functions can be assessed in a neutralization assay [41], potentially favoring some mAbs over others. Finally, the conversion of Elecsys U/mL to nM may be imprecise, as it is not specifically calibrated for SOT.

While these limitations highlight the importance of randomized clinical trials to evaluate mAb efficacy, the “Coronavirus Resistance Database” by Tzou et al., a living meta-analysis being updated as new data emerge, may still be used to guide the decision of which mAb to employ [24]. Regarding the currently predominant VOCs, Omicron BA.4/5, it shows a similarly reduced susceptibility between SOT and TIX/CIL, with 25-fold reduced neutralization. However, SOT shows improved neutralization of the BA.2.75 variant, with a 13-fold reduction in effectiveness compared to the same 25-fold reduction for TIX/CIL [24]. 

BIs occurred in 10% of our patients (Table 2 and Figure 3). While BIs occurred disproportionately more often after SOT PrEP (11/13), this is readily explained by the time point of most infections, which happened predominantly in March during the peak of the Omicron BA.2 wave in Austria, before TIX/CIL was being administered at our center. By comparison, during the same timeframe, over 33% of the Austrian population had confirmed infections with SARS-CoV-2 [42]. While the comparison to this group is limited due to various biases, it still highlights the widespread infections caused by the Omicron BA.1 and BA.2 waves in Austria in 2022. Furthermore, the disease course of BIs was mild: only one hospitalization occurred among the BIs, with the patient never requiring supplemental oxygen and being discharged after 2 days. Viral clearance (Ct Value > 30) occurred after a mean of 10 days, whereas previous studies in B-cell-depleted patients often showed significantly longer positivity, around 80 days [43], with frequent cases of prolonged and persistent disease [44,45,46,47].

The strengths of this study lie in the longitudinal comparison of two mAbs, with frequent follow-ups to assess quantitative antibody levels and incidence of BIs. Patient adherence was high with follow-ups available for all but one patient, with a median observation period of 93.9 days. The main limitations are the retrospective study design, the lack of a control group, and the absence of antibody neutralization assays.

In conclusion, our study highlights the potential role of SOT in SARS-CoV-2 PrEP for immunocompromised patients. Due to the fast-evolving nature of VOCs, SOT may present an additional tool, to be employed when the currently predominant strain is susceptible, as may be the case with Omicron BA.2.75. The antibody levels measured in our patients remained high even several months after infusion, overall incidence of breakthrough infections was low, and viral clearance was fast. However, larger, multi-center randomized trials are required to investigate the clinical effectiveness of SOT PrEP in preventing infection, severe disease, and death.

## Figures and Tables

**Figure 1 viruses-14-02278-f001:**
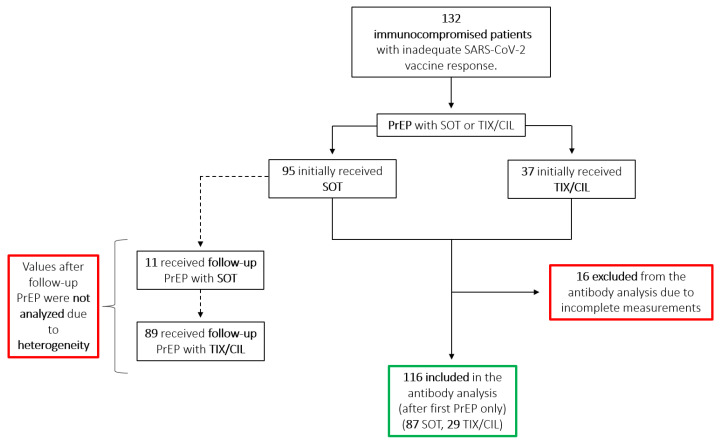
Flowchart showing the distribution of initial and follow-up PrEP among SOT and TIX/CIL. Antibody values resulting from follow-up PrEP were not analyzed due to the heterogeneity of mAb sequence and follow-up timing. The green border highlights included measurements; the red borders highlight excluded measurements. PrEP = Pre-exposure prophylaxis; SOT = sotrovimab; TIX/CIL = tixagevimab/cilgavimab; SARS-CoV-2 = severe acute respiratory syndrome coronavirus 2.

**Figure 2 viruses-14-02278-f002:**
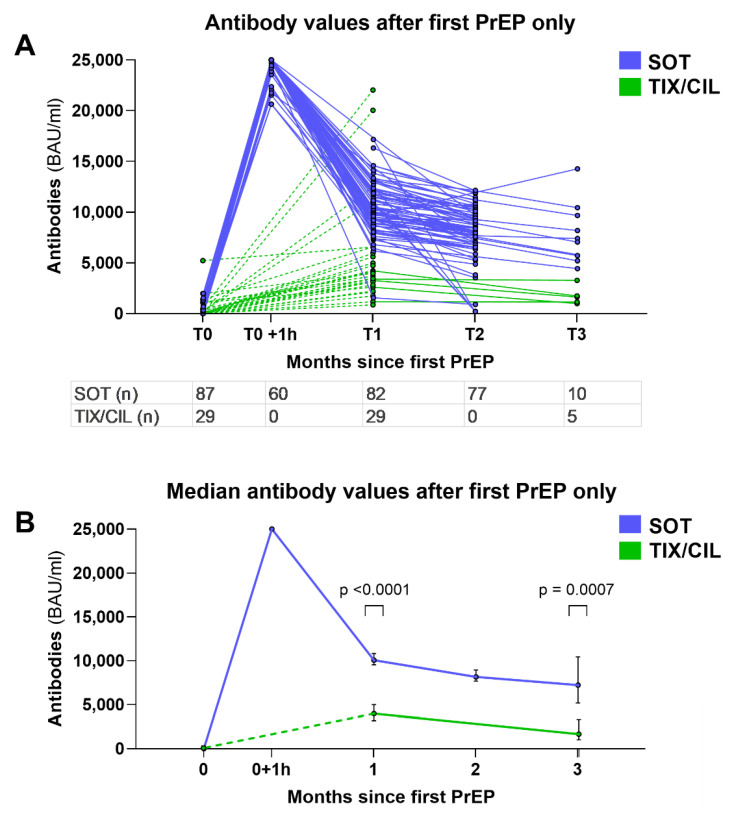
Graphs displaying the measured antibody values after first PrEP (before administration of any further PrEP), separated by time point. (**A**) shows all values after first PrEP, with SOT and TIX/CIL respectively. Each line represents a patient. The table shows the number of measurements available at each time point. (**B**) shows the median values of A. Error bars show 95% confidence interval. Mann–Whitney U test was used to compare the values at T1 and T3. T0 +1 h contains antibody values measured 1 h after SOT infusion, which could not be obtained for TIX/CIL due to delayed release after intramuscular injection. Dashed lines symbolize the inaccurate representation of pharmacokinetics in this case. BAU = binding antibody units; mL = milliliter; PrEP = pre-exposure prophylaxis; SOT = sotrovimab; TIX/CIL = tixagevimab/cilgavimab; T0–T3 = time points of measurements, roughly equal to months since first PrEP.

**Figure 3 viruses-14-02278-f003:**
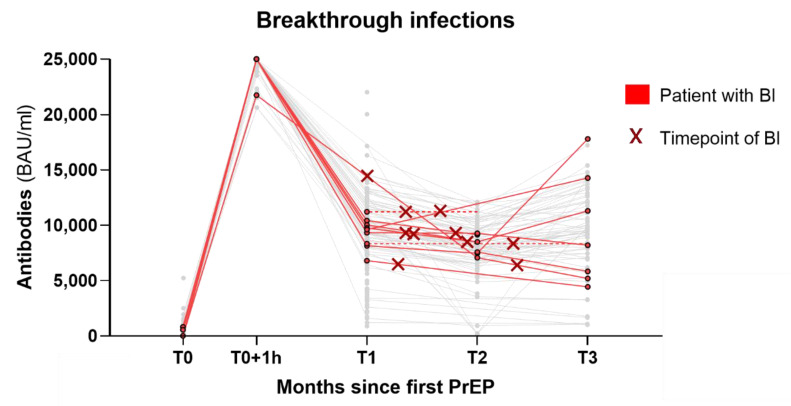
Graph of the BIs that occurred. Each red line represents a patient with BI while all patients without BI are greyed out. The time point of BI is highlighted with a red cross. The dotted lines are projections of the last available values for patients with missing follow-up measurements. Three patients with BI are not represented in this graph due to missing antibody values. BI = breakthrough infection; PrEP = pre-exposure prophylaxis; BAU = binding antibody units; mL = milliliter; T0–T3 = time points of measurements, roughly equal to months since first PrEP.

**Table 1 viruses-14-02278-t001:** Patient characteristics.

Demographic Parameters (*n* = 116)	
Age (years)	59.6 (±15.1)
Male/Female (% Male)	53/63 (46%)
Weight (kg)	75.5(±15.7)
Number of SARS-CoV-2 vaccinations received before first PrEP	3.7 (±0.9)
**Underlying Disease**	
Hematologic malignancy	49 (42%)
Autoimmune disease	27 (23%)
Multiple sclerosis	21 (18%)
Immunodeficiency	10 (9%)
Organ transplantation	9 (8%)
**Immunosuppressive Medication**	
Anti-CD20	39 (34%)
Fingolimod	14 (12%)
Tacrolimus/MMF	12 (10%)
Other immunosuppressive therapy	12 (10%)
Other oncological therapy	7 (6%)
Unknown	7 (6%)
None	25 (22%)
**COVID-19 PrEP**	
Median antibody-level before first infusion (BAU/mL)	9 (IQR = 250)
First PrEP with SOT	87 (75%)
First PrEP with TIX/CIL	29 (25%)
Follow-up PrEP with SOT	7
Follow-up PrEP with TIX/CIL	83
Median duration of follow-up (days since first PrEP)	93.9 (± 43.4)

Age, weight, number of vaccinations, and follow-up duration are given as mean values with standard deviation. COVID = coronavirus disease 19; PrEP = pre-exposure prophylaxis; Kg = kilogram; BAU = binding antibody units; mL = milliliter; IQR = interquartile range; SOT = sotrovimab; TIX/CIL = tixagevimab/cilgavimab; anti-CD20 = antibodies against the CD20 receptor; MMF = mycophenolate mofetil.

**Table 2 viruses-14-02278-t002:** Characteristics of patients with breakthrough infection.

Patients with Breakthrough Infection (*n* = 13)	
Age (years)	60.2 (±13.9)
Male/Female %	9/13 (69%)
**Rate of breakthrough by underlying disease**	
Hematologic malignancy	8/58 (13.8%)
Multiple sclerosis	3/24 (12.5%)
Immunodeficiency	1/11 (9%)
Organ transplantation	1/9 (11%)
Autoimmune disease	0/30
**Rate of breakthrough by immunosuppressive medication**	
Other oncological therapy	4/9 (44.4%)
Anti-CD20	3/46 (6.5%)
Fingolimod	2/15 (13.3%)
Unknown	2/9 (22.2%)
Tacrolimus/MMF	1/13 (7.7%)
None	1/28 (3.6%)
Other immunosuppressive therapy	0/12
**Breakthrough infection**	
Incidence	13/132 (10%)
Median days since last PrEP	40.0 (IQR = 34.5)
Last PrEP with SOT	11/13
Last PrEP with TIX/CIL	2/13
Hospitalization	1/13
Supplemental oxygen	0/13
Median duration of symptoms (days)	4.0 (IQR = 8)
Median duration until Ct-value > 30 (days)	10.0 (IQR = 16.5)

Age is given as mean with standard deviation. PrEP = pre-exposure prophylaxis; Kg = kilogram; IQR = interquartile range; SOT = sotrovimab; TIX/CIL = tixagevimab/cilgavimab; MMF = mycophenolate mofetil; anti-CD20 = antibodies against the CD20 receptor; Ct-value = cycle threshold value.

## Data Availability

The data referred to during the study are available from the corresponding author on reasonable request.

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
