# Peer review of "SARS-CoV-2 Pre-Exposure Prophylaxis with Sotrovimab and Tixagevimab/Cilgavimab in Immunocompromised Patients—A Single-Center Experience"

_viruses, 2022, doi:10.3390/v14102278_

Round 1

Reviewer 1 Report

This is a single-center, retrospective and interesting study aimed at evaluating antibody levels of two monoclonals (sotrovimab and tixagevimab/cilgavimab) at several times after their administration as pre-exposure prophylaxis.

The main problems are

-Unclear and confusing distribution of patients between sotrovimab and tixagevimab/cilgavimab groups makes the analysis of antibody levels unclear.

-Unclear number of doses in the sotrovimab group and their effect on antibody level.

-What is the study's objective: to describe the antibody levels at T1-T3 in those patients with 1 dose of sotrovimab vs one dose of tixagevimab/cilgavimab, or to compare a dosing of 2 doses of sotrovimab with one dose of tixagevimab/cilgavimab at T3?

-Unclear effect of each monoclonal on breakthrough infections for the same reasons (mix antibodies administrations, number of doses unknown, …).

a) Abstract

1.-Change units to BAU/ml, and for the rest of the manuscript.

2.-It is not necessary to capitalise tixagevimab/cilgavimab and sotrovimab. Both are active substances, not commercial names.

b) Introduction

1.-Please, change the abbreviation for sotrovimab. “SOT” usually is used for solid organ transplants and may induce confusion.

c) Methods

1.- Inadequate vaccination response was defined as antibody titers below 250 U/ml. But, after how many minimum doses of vaccine? If it is after 3 doses, it is ok, but if it is after only one, it is clearly too premature point to define failure. Please, clarify and specify how many doses the patients received before PrEp.

2.-Dosing of the different monoclonals and the number of patients in each subgroup.

-The dosing of tixagevimab/cilgavimab and sotrovimab is different. Tixagevimab/cilgavimab was given once, but sotrovimab was given every 2 months.

- Looking at figure 1 is not clear how many received only sotrovimab (one or more doses), how many received only tixagevimab/cilgavimab and how many received the combination (first sotrovimab and later tixagevimab/cilgavimab)

-What do you mean by “follow-up PrEp” in figure 1?

-Moreover, in table 1, in the Sotrovimab arm (95 patients with initial dose of  this antibody), 11 received follow-up PrEP with sotrovimab, and 89 received follow-up PrEp with tixagevimab/cilgavimab, that sum 100. It means that there were some patients that initially received sotrovimab, next follow-up PrEp with sotrovimab and finally follow-up PrEp with tixagevimab/cilgavimab. This makes it difficult to know how many patients received what.

3.-For the analysis of antibody levels, please make a simpler description of the patients: those who received only one dose of sotrovimab compared to those that received only one dose of tixagevimab/cilgavimab. Detail the characteristics of these patients in a table.

4.- 116 patients analysed for antibody levels in figure  2 B-C, the time frame includes 3 months after PrEP. As sotrovimab was given every 2 months, how many of the 87 patients in the sotrovimab group received only one dose? Patients receiving more than 1 dose could increase the levels of T2 and T3 compared to one dose of tixagevimab/cilgavimab. This need to be clarified.

d)Results

1.-Initially, there were 132 patients, but 16 were excluded due to incomplete data. In the end, there were 116 patients. The number of 132 only is interesting for breakthrough infections.

-We need another table focused on the 116 patients analyzable for antibody levels.

-More detail about patients would be welcome:

            -Type of hematologic malignancy and phase of the disease

            -Type o organ transplantation

2.-Time points for antibody levels

-T1 is a heterogeneous time point, that goes from day 0 to day 44, which could introduce an important bias in the antibodies levels if you compared one taken in 24h after administration to another taken 44 days later.

-T4 and T5 are not useful as only 5 patients with antibodies levels after 3 months of monoclonal administration, and only for tixagevimab/cilgavimab.

Please eliminate T4 and T5 as they don't add useful information.

-We need another table with the following data:

-How many patients did you have for each antibody at T1, T2 and T3?

-What was the median day (and range) for T1, T2, and T3 for sotrovimab and tixagevimab/cilgavimab antibody levels?

3.-Despite describing the time points T1, T2, …. You give the results by months in the text (lines 153-156) but with time points in figures 2 B-C. If you finally give the results by month, why did you describe time points?

4.-Moreover, when you say 1, 2 and 3 months, do this mean exactly 1, 2 and 3 months, or are an approximation like time points? Please use consistent terminology across the manuscript.

5.-In figure 2 C there is no P value for T2. Why?

6.-Levels of antibodies

            -Lines 153-155: Please clarify to which patients correspond the data. To the 116 or 132 patients?
“Median antibody values 1, 2 and 3 months after SOT were 10058, 8160 and 7235 U/ml 153, respectively. Median antibody values 1 and 3 months after TIX/CIL were 3965 (p < 0.0001) 154 and 1647 (p= 0.0007) U/ml, respectively”

7.-Breakthrough infections

-There were 13 breakthrough infections, but in figure 3 there are only 10. Explain or correct.

-Table 2

                        -Please express the percentages based on the overall number of patients by underlying condition and Immunosuppressive medication: hematologic malignancy: 8/58 (13.7%), multiple sclerosis (3/24= 12.5%), and so on. In this manner, the rate by Underlying Disease conditions is quite similar, for example.

            -If you analyse BI by type of monoclonal, the rate was 11/95 (11.5%) for sotrovimab and 2/37 (5.4%) for tixagevimab/cilgavimab. Nonetheless, as some sotrovimab patients receive one or more doses or tixagevimab/cilgavimab, the protective effect of each monoclonal is challenging to assess.

            In table 2, it would be helpful to add the type and doses of monoclonals received, not only the last one.

e)Discussion

1.-Lines 248-249: “BIs occurred in 10% of our patients (Table 1 & Figure 3). While BIs occurred disproportionately more often after SOT PrEP (11/13) …” 11/13 = 84.6%.

When the number of patients that received sotrovimab and tixagevimab/cilgavimab is quite different, the percentage of BI cases with each monoclonal is misleading.   If you analyse BI by type of monoclonal, the rate was 11/95 (11.5%) for sotrovimab and 2/37 (5.4%) for tixagevimab/cilgavimab

Reviewer 2 Report

This is a retrospective, single-center chart review study of 132 immunocompromised patients (with hematologic malignancy, autoimmune disease, MS, and other immunocompromising conditions) with insufficient humoral responses to SARS-CoV-2 vaccination (defined as titers < 250 U/mL in the Roche Elecsys assay) who received sotrovimab 500mg IV off-label (planned to be given every 2 months, N = 95) for COVID PrEP, switching in March 2022 to tixagevimab/cilgavimab (N = 37) instead with Omicron BA.2 spreading. The authors characterized the levels of sotrovimab after infusion, checked at various times post-PrEP, comparing titers over 3 months vs. tix/cil and finding a 10% (13/132) rate of breakthrough infections with relatively mild clinical disease. Overall, it is well-written and a relatively unique piece, as sotrovimab was not approved for a PrEP indication. In addition to the limitations the authors mention (lack of a control group, and the absence of Ab neutralization assays), other limitations that should probably be mentioned are the retrospective design of the study, with data not collected at uniform time points post-infusion (it appears that values collected from patients within a certain interval were lumped together, making the figure a little misleading), with likely missing data (how were missing data points handled?) and a lack of systematic testing/surveillance for breakthrough infections in the study cohort, with the possibility of missing patients with milder infections who did not seek testing. It is somewhat interesting that there was no difference in antibody trajectory (based on the figure) in patients with BI and patients without BI, although a formal comparison using mixed effects models would be interesting. I'm not sure of the clinical significance of the observation that the antibody levels using the Elecsys assay were higher in patients receiving SOT than TIX/CIL, with correlates of protection still being undefined, but I think the authors do not overstate the importance of this finding. Table 2 is likely unnecessary, with the small. number of breakthrough infections and any significant differences in these parameters in the infected vs. non-BI groups should probably just be noted in the text. 
